# Raising Epidemiological Awareness: Assessment of Measles/MMR Susceptibility in Highly Vaccinated Clusters within the Hungarian and Croatian Population—A Sero-Surveillance Analysis

**DOI:** 10.3390/vaccines12050486

**Published:** 2024-05-01

**Authors:** Dávid Szinger, Timea Berki, Ines Drenjančević, Senka Samardzic, Marija Zelić, Magdalena Sikora, Arlen Požgain, Ákos Markovics, Nelli Farkas, Péter Németh, Katalin Böröcz

**Affiliations:** 1Department of Immunology and Biotechnology, Clinical Center, Medical School, University of Pécs, 7624 Pécs, Hungary; szinger.david@pte.hu (D.S.); berki.timea@pte.hu (T.B.); nemeth.peter@pte.hu (P.N.); 2Department of Physiology and Immunology, Faculty of Medicine Osijek, Josip Juraj Strossmayer University of Osijek, 31000 Osijek, Croatia; idrenjancevic@mefos.hr; 3Scientific Centre for Excellence for Personalized Health Care, Josip Juraj Strossmayer University of Osijek, 31000 Osijek, Croatia; 4Department of Public Health, Teaching Institute of Public Health for The Osijek-Baranja County, 31000 Osijek, Croatia; senka.002@gmail.com (S.S.); marija.zelic@zzjzosijek.hr (M.Z.); magdalena.sikora9@gmail.com (M.S.); arlenpozgain@gmail.com (A.P.); 5Department of Microbiology, Parasitology and Clinical Laboratory Diagnostics, Medical Faculty of Osijek, Josip Juraj Strossmayer University of Osijek, 31000 Osijek, Croatia; 6Department of General and Physical Chemistry, Faculty of Natural Sciences, University of Pécs, 7622 Pécs, Hungary; makos007@gamma.ttk.pte.hu; 7Department of Bioanalysis, Medical School, University of Pécs, Szigeti u. 12, 7643 Pécs, Hungary; nelli.farkas@aok.pte.hu

**Keywords:** vaccine, measles, mumps, rubella, MMR, sero-epidemiology, age-stratified, risk group, disruption, vaccination effort, suboptimal, seropositivity, pandemic

## Abstract

Perceptions of the complete eradication of vaccine-preventable diseases such as measles, mumps, and rubella (MMR) may foster complacency and compromise vaccination efforts. Decreased measles vaccination rates during the COVID-19 pandemic have heightened the risk of outbreaks, even in adequately vaccinated populations. To address this, we have aligned with ECDC recommendations, leveraging previous cross-border sero-epidemiological assessments between Pécs, Hungary, and Osijek, Croatia, to identify latent risk groups and uncover potential parallels between our nations. Testing 2680 Hungarian and 1764 Croatian serum samples for anti-MMR IgG via ELISAs revealed anti-measles seropositivity ratios below expectations in Croatian cohorts aged ~20–30 (75.7%), ~30–40 (77.5%) and ~40–50 years (73.3%). Similarly, Hungarian samples also showed suboptimal seropositivity ratios in the ~30–40 (80.9%) and ~40–50 (87.3%) age groups. Considering mumps- and rubella-associated seropositivity trends, in both examined populations, individuals aged ~30–50 years exhibited the highest vulnerability. Additionally, we noted congruent seropositivity trends across both countries, despite distinct immunization and epidemiological contexts. Therefore, we propose expanding research to encompass the intricate dynamics of vaccination, including waning long-term immunity. This understanding could facilitate targeted interventions and bolster public awareness. Our findings underscore persistent challenges in attaining robust immunity against measles despite vaccination endeavors.

## 1. Introduction

Despite the perception of vaccine-preventable diseases being eradicated in modern society, it is crucial to recognize the persistent importance of maintaining high vaccine coverage rates, ensuring vaccine quality and potency, monitoring population immunity levels and implementing supplementary vaccination measures as warranted. This entails a proactive stance, as relying solely on past successes may foster complacency and undermine efforts to sustain immunity against measles and other vaccine-preventable diseases.

Addressing the challenges associated with the shortcomings in measles vaccine efficacy and inadequate seroconversion demands a comprehensive socio-epidemiological approach, necessitating an understanding of the multifaceted influences of political, economic, cultural and social factors on immune-serological protection outcomes [1,2,3,4,5,6,7,8,9].

The COVID-19 pandemic has underscored significant socioeconomic disparities in health outcomes and access to healthcare. Additionally, it has precipitated a notable and protracted reduction in childhood vaccination rates, which is estimated to be the most substantial decline observed in approximately three decades. This alarming trend has led to an estimated shortfall of vaccinations for approximately 25 million children globally [10,11,12,13]. Socioeconomic inequalities (SESs) constitute pivotal determinants influencing vaccine uptake [12,14], with indicators encompassing income, occupation, education and, occasionally, place of residence [12,15]. Notably, socioeconomic differentials in vaccination uptake exhibit spatial and temporal variability, spanning between-group and within-group distinctions such as ethnicity or gender, evolving alongside changes in healthcare systems and policies over time and fluctuating across different administrative contexts and countries. Moreover, the influence of socioeconomic factors on vaccination patterns extends across various vaccines, including MMR, contributing to nuanced dynamics in vaccine acceptance and utilization [12,16,17,18,19,20].

Additionally, human migration carries substantial implications for national economies, healthcare systems and social cohesion [1,2,3,4,5,6,7,8,9]. Factors such as regional conflicts, economic crises and global warming contribute to large-scale migrations. A key concern is the potential for increased risk of vaccine-preventable disease outbreaks, such as measles, stemming from the mass movement of individuals, often from regions with compromised healthcare systems, exacerbated by the impacts of the COVID-19 pandemic [3,7].

Furthermore, a multitude of human immune biological variables must be taken into account [21,22,23,24,25,26], including but not limited to declining immunity over time, incomplete vaccination regimens, variable vaccine efficacy across viral strains, vaccine storage and handling protocols, individual immunological responses, interference from maternal antibodies and occurrences of vaccine breakthrough infections.

The measles, mumps, and rubella (MMR) vaccine demonstrates high efficacy, with the standard two-dose regimen conferring protective immunity to approximately 99% of recipients [27,28,29], and it is provided free of charge within the examined countries.

Nevertheless, measles epidemics represent a longstanding challenge, with the measles virus afflicting humanity for centuries [30,31,32]. However, the current surge in cases diverges from historical outbreaks characterized by periodic fluctuations prior to widespread vaccination efforts. Notably, these contemporary outbreaks also occur within populations boasting high vaccination coverage [3,8,24,25,26,27,28,29,30,31,32,33,34,35], leading to potential confusion among the public and presenting challenges for healthcare professionals, epidemiologists, and clinical microbiologists [33]. While the prevalence of vaccination among individuals significantly exceeds that of those who are unvaccinated, there appears to be a subset within the well-immunized cohort that may remain susceptible to unexpected infections, potentially introduced through imported cases such as unvaccinated travelers. Consequently, even a minor proportion of vaccinated individuals, encompassing also the aforementioned immunization gaps contracting the virus, could contribute substantially to the overall case count [32,33,34,35].

The surge in cases intensified toward the end of 2023 and has persisted into 2024. Figure 1 illustrates the alarmingly increasing case numbers between 2022 and 2023, as featured by the European Centre for Disease Prevention and Control (ECDC) Surveillance Atlas for Infectious Diseases [36]. It is anticipated that measles cases will continue to rise across the EU/EEA in the coming months. This trend can be attributed primarily to suboptimal vaccination coverage for measles-containing vaccines (MCVs) in specific EU/EEA countries, heightened susceptibility to importation from areas with widespread circulation of the virus and the convergence of the upcoming months with the seasonal peak of measles transmission [37].

The WHO European region—encompassing a considerably larger geographic scope compared to the territory of the ECDC Europe—witnessed a surge in measles cases in 2023, with more than 30,000 cases documented across 40 of the 53 countries, resulting in 21,000 hospitalizations. This escalation has continued into 2024 (Figure 2). Measles infections have impacted individuals of all age groups, with notable disparities in age distribution observed among different countries. Approximately two out of every five reported measles cases involve children under the age of five [38]. Austria and Romania rank among the ten most affected countries, as identified by the WHO Regional Office for Europe, although Kazakhstan, Kyrgyzstan, and Armenia exhibit the highest incidence rates [39]. The upsurge in measles cases is largely attributed to the decline in vaccination coverage during the COVID-19 pandemic from 2020 to 2022. This decline resulted in a significant rise in the number of individuals who were either unvaccinated or incompletely vaccinated, both within the European region and globally [38,40].

In December 2023, the US Centers for Disease Control and Prevention (CDC) released the updated 2024 Advisory Committee on Immunization Practices (ACIP) Adult Immunization Schedule. These guidelines were introduced during a concerning rise in vaccine-preventable viral infections, such as SARS-CoV-2, and the resurgence of measles, once thought to be eradicated. This increase is attributed to vaccine hesitancy and noncompliance [41]. Recent data from the CDC present alarming evidence confirming the ongoing problem. As of 11 April 2024, a cumulative total of 121 measles cases had been reported across 18 jurisdictions: Arizona, California, Florida, Georgia, Illinois, Indiana, Louisiana, Maryland, Michigan, Minnesota, Missouri, New Jersey, New York City, New York State, Ohio, Pennsylvania, Virginia, and Washington. There have been seven outbreaks (defined as 3 or more related cases) reported in 2024, and 71% of cases (86 of 121) are outbreak-associated [42]. The re-emergence of measles virus infections, previously under control in Western countries due to the measles, mumps, and rubella (MMR) vaccine, underscores the public health risks associated with insufficient efforts to emphasize the importance of vaccination adherence [41,42,43,44,45,46,47,48,49,50,51,52].

Moreover, in January 2024, the WHO Region of the Americas issued an epidemiological alert concerning measles, urging countries in the region to persist in their efforts to improve and sustain sufficient vaccination coverage against measles, rubella, and mumps [53]. According to this ‘Epidemiological Alert’ (PAHO/WHO; January 2024, Washington, DC, USA), in light of the persistently low coverage rates of MMR1 and MMR2 vaccines, the escalating global incidence of measles, and the emergence of imported cases within the Americas, the Pan American Health Organization/World Health Organization (PAHO/WHO) advocates for Member States to enhance vaccination coverage against measles, rubella, and mumps. A focus on vaccination, robust surveillance systems and swift response capabilities constitutes fundamental strategies to sustainably control the spread of these viruses [53].

Regarding preventive intervention measures in Europe, the European Centre for Disease Prevention and Control (ECDC) advises public health authorities within the European Union (EU) and European Economic Area (EEA) to prioritize the following fundamental strategies (Figure 3): addressing immunity gaps and achieving high vaccination coverage for measles-containing vaccines, enhancing the quality of surveillance systems, strengthening public health capacities, particularly in terms of outbreak control, and improving compliance in vulnerable settings to enhance acceptance and adherence to vaccination efforts [37].

Accordingly, the main scope of the present immune-serological analysis is to address population-wide immunity gaps against the presumably eradicated vaccine-preventable diseases measles, mumps and rubella. While focusing on medicine and epidemiology, broader socio-economic and geopolitical factors influencing herd immunity are crucial to consider. With Ukraine’s significant population and history of measles outbreaks, disruptions in healthcare and vaccination schedules heighten epidemiological risks. Experts warn of potential measles outbreaks in Ukraine, stressing the need for proactive measures [54,55,56,57,58,59,60,61]. Additionally, the CDC cautions that infectious diseases can rapidly spread, reaching major urban centers worldwide within 36 h. Measles, with its relatively long incubation and latency period, poses significant risks. Decreases in measles vaccination rates during the COVID-19 pandemic have raised the global outbreak risk, with over 61 million doses of measles-containing vaccine postponed or omitted between 2020 and 2022 due to COVID-19 disruptions [37,56,61,62,63,64,65].

Hence, we conducted an age-stratified study, synthesizing data from a cross-border collaboration with Osijek, Croatia. Leveraging former sero-epidemiological assessments of anti-measles (and subsequently anti-MMR) humoral antibody levels (IgG), subsequently transformed into seropositivity ratios, our aim was to elucidate age-specific epidemiological patterns, identify high-risk cohorts and discern potential analogies through a comparative analysis between the cross-border regions of Hungary and Croatia. Our model stands as a robust and equitable representation of the South–Western European population from an epidemiological standpoint, given the historically consistent demographic characteristics over recent decades. Our principal objectives encompassed the delineation of age-specific risk profiles and the exploration of age-related epidemiological dynamics within adjacent nations. We suggest that the herein presented analysis might help in the assessment of targeted intervention feasibility tailored to specific age strata amidst potential disease importation scenarios and the overarching endeavor of enhancing societal awareness.

## 2. Materials and Methods

### 2.1. Human Serum Samples

Within the framework of a double-centered cross-border cooperation with Osijek, Croatia, we evaluated a total of 2680 residual anonymous serum samples from Hungary (Department of Laboratory Medicine, Department of Immunology and Biotechnology, Clinical Centre, Medical School, University of Pécs, Hungary) and 1764 residual anonymous serum samples received from Osijek, Croatia (Scientific Centre for Excellence for Personalized Health Care, Josip Juraj Strossmayer University of Osijek). Detailed sample numbers are represented in Table 1.

### 2.2. ImmunoSerological Meassurment of Human Serum Samples

As outlined in our previously published research on immunoassay scaling, setup and refinement [66,67], our laboratory executed triple measles, mumps, and rubella (MMR) enzyme-linked immunosorbent assays (ELISAs) utilizing the robotic functionalities of the Siemens BEP 2000 Advance System (Siemens/Dade Behring, Marburg, Germany). During the development of our in-house ELISAs, commercially available antigen preparations were utilized, comprising the measles Edmonston strain cultured in Vero cells (PIP013 Bio-Rad, Hercules, CA, USA), the mumps Enders strain cultured in BSC-1 cells (PIP014 Bio-Rad), and the rubella HPV-77 strain cultured in Vero cells (PIP044 Bio-Rad). The calibration of internal standards adhered to established international standards, recognized as ‘gold standards’, including the 3rd WHO International Standard for Anti-Measles (NIBSC 97/648), Anti-Rubella Immunoglobulin 1st WHO International Human Standard (NIBSC RUBI-1-94), and Anti-Mumps Quality Control Reagent Sample 1 (NIBSC 15/B664).

Prior to the automated execution of the assays, manual specimen pre-analytics and sample predilution were performed. As described earlier [66,67], self-developed MMR ELISAs underwent optimization utilizing a range of commercially available reference assays from various providers (Novalisa, Sekisui -Virotech, Immunolab, Serion, Euroimmun, Siemens Enzygnost, Vircell, Novatec, DiaPro, ORGENTEC Alegria^®^ Test Strips). For an independent reference method, a population-based validation of the in-house MMR ELISAs’ indirect immunofluorescence (EUROIMMUN Medizinische Labordiagnostika AG. Lübeck, Germany) was employed, also confirmed by an independent laboratory (National Centre for Epidemiology, Department of Virology, Budapest, Hungary). Additionally, as an in-house reference method, the results were verified using monoclonal anti-viral antibody-based sandwich ELISAs. Optimization experiments were conducted to enhance the signal-to-noise ratio and minimize nonspecific background, guided by the concordance with reference tests [66,67]. Assay precision and specific assay characteristics across various variables have been comprehensively outlined in our prior publications [66,67]. As outlined in the prior literature [66,67], the determination of the threshold was achieved via an analysis of Receiver Operating Characteristics (ROCs), employing the Area Under the Curve (AUC) method, coupled with the application of Youden’s J equation. Subsequently, these normative reference cut-off values were refined according to the findings of reference commercial tests. Extinction values were transformed into quantitative data by fitting sigmoidal dose–response curves to the dilution points of the standards. Appendix A provides a schematic representation of the fundamental steps of the assay protocol.

### 2.3. Methods of Result Evaluation

To conduct a rigorous analysis of age-stratified sero-epidemiological data intended to monitor variations in humoral immunity between two distinct nations, seropositivity ratios were determined using the formula seropositivity = (number of positive samples per age group/total number of samples per age group) × 100. These ratios were then graphically presented via dot plots for a clear and concise visualization of the results. OriginLab data analysis and graphing software was employed to ensure accuracy and efficiency in the graphical presentation of the findings.

To comprehensively analyze the data from a statistical perspective, we implemented the Clopper–Pearson exact binomial confidence interval as a statistical method in order to calculate the confidence intervals for proportions regarding our binomial data (positive or negative seropositivity outcomes) for both Croatia and Hungary simultaneously. The absence of overlap in the confidence intervals (CI 95%) was interpreted as indicative of a statistically significant difference between the respective age groups. Recognizing the potential variability stemming from varying case numbers, we opted against presenting p-values as a measure of statistical significance. (For multiple statistical tests, such as comparisons across various variables, False Discovery Rate Correction may be necessary to overcome the increased likelihood of observing a ‘significant’ result by random chance alone.)

## 3. Results

The observations illustrated in Figure 4a–c (based on Table 2) regarding anti-measles, mumps, and rubella (MMR) seropositivity ratios within the Croatian and Hungarian populations elicit significant epidemiological concerns, particularly with respect to measles (Figure 4a). Given the widely cited basic reproduction number (R0) for measles, typically falling within the range of 12–18, which denotes the average number of secondary infections generated by a single infectious individual within a wholly susceptible population [29,31,32,68], and acknowledging that achieving herd immunity for measles necessitates vaccination coverage of ≥95%, alongside an expected seroconversion rate of ≥95–98% [1,9,12,13,14,16,18,20,40,51,55,56,57,58,59,60,61,62,63,64,65,66,67,68], the specific findings delineated in the age-stratified clusters present in Figure 4a–c are less than reassuring. Despite concerted immunization efforts, suboptimal anti-measles seropositivity ratios were detected in multiple age clusters.

The highest vulnerability within the Croatian cohorts was recorded in the age groups 20–30, 30–40, and 40–50 years, demonstrating seropositivity ratios of 75.7%, 77.5%, and 73.3%, respectively. Similarly, among the Hungarian samples, suboptimal seropositivity ratios were observed within the age clusters of approximately 30–40 and 40–50 years, with 80.9% and 87.3% seropositivity ratios, respectively.

Moreover, regarding mumps- and rubella-associated seropositivity ratios, individuals within the age range of approximately 30–50 years in both examined nations exhibited heightened vulnerability (Figure 4b,c). The conspicuous deficit in humoral antibody titers and subsequent existence of seropositivity gaps within this age cohort prompt inquiries into vaccine efficacy [24,30,37,69,70,71,72,73,74,75,76,77,78,79,80,81,82,83]. These findings echo previous studies [35,66,67,69,70,71] and data in the literature [72,73,74,75,76,77,78,79], indicating persistent challenges in achieving sufficient immunity against measles despite vaccination efforts.

The highest seropositivity ratios in both adjacent nations for measles, mumps and rubella were observed among individuals aged 50 years and older (Figure 4a–c, based on Table 2). This demographic cluster likely experienced natural infections either during the early stages of vaccination or before widespread vaccination efforts were implemented. Our observation aligns with the established literature suggesting that immune protection resulting from wild-type infection elicits a more durable and robust immune response compared to vaccine-induced immunity [35,66,69,70,80,81,82,83,84,85].

Furthermore, it is noteworthy to observe relative similarities in the trends and overlaps of seropositivity patterns between the two countries (Figure 4a–c), despite known disparities in vaccination protocol schedules, immunization inocula, geopolitical histories and consequent healthcare system statuses and availability. This finding is also consistent with prior observations [35].

Furthermore, age-related disparities are also detectable at a statistical level (Figure 5a–c). Regarding measles (Figure 5a), the analyses of both the Hungarian and Croatian cohorts reveal notable distinctions between earlier vaccination eras (age > 50 years) and younger age groups (20–30, 30–40 and 40–50 years) of vaccinated individuals (age < 50 years), as discerned from nonoverlapping confidence intervals (Figure 5a–c, Appendix A). These findings substantiate our hypothesis: the initial recipients of targeted vaccination may have still encountered residual virus circulation during the program’s inception, while subsequent age groups might have experienced vaccine failure and/or immune senescence due to a lack of natural boosting (i.e., disrupted virus circulation). Notably, the youngest vaccine recipients (10–20 years) in both countries exhibit sufficiently high seropositivity rates comparable to the elder clusters.

In the analysis of Hungarian mumps (Figure 5b) data, statistically significant differences are most pronounced among age groups 30–40, 50–60 and 60–70. Intriguingly, in the Croatian samples (Figure 5b), no statistically significant nonoverlaps were observed.

Regarding rubella (Figure 5c), significant statistical disparities were found exclusively in the Croatian samples, with the most notable contrast observed between age groups 30–40 and 50–60 years when considering nonoverlapping confidence intervals.

## 4. Discussion

In developed European regions with high vaccination rates and generally favorable compliance, elucidating gaps in humoral protection presents a significant challenge in addressing susceptibility to infection. These gaps are closely associated with instances of vaccine failure, which can be attributed to various factors, including the age of the vaccine recipient at the time of vaccination, the vaccination regimen, the immunogenicity of the vaccine strain and regional demographic characteristics. Primary vaccine failure, characterized by individuals failing to mount an adequate immune response following vaccination, and secondary vaccine failure, occurring when individuals previously exhibiting serological conversion following vaccination experience measles infection, contribute to the complexity of this issue [72,73,76,77,78,79,86,87,88].

Aligned with the directives set forth by the ECDC, our current article highlights key points emphasized in the February 2024 ‘Threat Assessment Brief’. The emphasis of our present analysis lies in addressing deficiencies in humoral immunity against measles, mumps and rubella (MMR), specifically focusing on anti-viral IgG-derived seropositivity ratios. Additionally, we suggest that identifying vulnerable population groups may facilitate efforts to mitigate the spread of the virus in scenarios involving sudden or unforeseen rises in potentially late-diagnosed imported cases. Despite vaccination coverage exceeding 95% with both doses of measles-containing vaccines (MCVs) in the examined countries (Hungary, Croatia), historical epidemiological data reveal instances of major epidemics occurring even after the implementation of mandatory and cost-free vaccinations [38,72,73,74,75,76,77,79,88,89]. This suggests the presence of potential imperfections or gaps in the epidemiological protective measures. Additionally, the basic reproduction number (R0) for measles is notably high, estimated to be between 12 and 18. This means that, on average, each person infected with measles could potentially transmit the virus to 12–18 others in a completely susceptible population Achieving herd immunity for measles necessitates vaccination coverage of at least 95%, coupled with an anticipated seroconversion rate of 95–98% [1,9,13,14,20,22,23,30,40,51,55,56,57,58,59,60,61,62,63,64]. However, the specific findings for the age clusters depicted in the results section (Figure 4 and Figure 5) are not reassuring, indicating a significant disparity between the observed seropositivity rates and the thresholds required for herd immunity.

In order to provide a comprehensive discussion of the age-cluster-related seropositivity analysis presented in the results section, it is pertinent to contextualize the vaccination schedules of the compared countries from an epidemiological perspective. Table 3 and Table 4 (Table 3: Hungary; Table 4: Croatia) provide a comprehensive overview of the modifications made to vaccination schedules, including the timing of primary and booster immunizations, targeted demographics, administration techniques, and the types and formulations of the administered vaccines. By delineating age group boundaries within the aforementioned tables, connections between vaccination and epidemiological trends become traceable and interpretable.

Despite the potential limitations in humoral protection levels indicated by the presented results, it is noteworthy that both Croatia and Hungary adhere to established and well-tested vaccination protocols. These protocols are based on the fundamental principle of mandatory, easily accessible, and cost-free MMR immunizations. In Croatia, children are vaccinated at 12 months of age and again at 6–7 years (grade 1 students) [90], while in Hungary, children receive the MMR immunization at 15 months and again at 11 years (as part of the routine school-based vaccination schedule in 6th grade) [90,91].

Our result analysis shows that in the Croatian cohort aged ~40–50 years, encompassing individuals born between 1983 and 1973 (±1 year), despite undergoing systematic immunization initiatives with concerted endeavors aimed at achieving comprehensive coverage on multiple occasions (Table 4), the levels of humoral anti-measles antibodies tested fall notably below expectations. Considering the age range of this cohort and adhering to the principle that earlier years of immunization protocol implementation are more likely to result in suboptimal seroconversion outcomes due to the perceived fledgling state of the system, our focus should be directed towards the early years of implementation.

As per the vaccination schedule delineated in Table 4, the primary vaccination was administered to children at the age of one, with subsequent inclusion of the rubella component. In 1974, the mumps component of the vaccine was also incorporated. During 1975, children older than one year adhering to a consistent vaccination regimen were slated for their initial vaccination dose. Furthermore, children born in 1973 eligible for targeted vaccination campaigns, excluding those awaiting their third DTaP dose, were designated to receive their first vaccination. Additionally, unvaccinated children over one year of age attending preschools were also scheduled for their initial vaccination. Moreover, children born in 1971 and those entering first grade during the 1974/75 academic year were included in the initial vaccination plan.

Based on the aforementioned data, it appears highly probable that insufficient vaccination coverage was not the primary issue. Commonly cited traditional factors contributing to vaccine insufficiency include waning immunity, incomplete vaccination series, variability in vaccine effectiveness and suboptimal conditions during distribution and administration. Consequently, attention may be redirected towards the composition of the vaccine itself. Historical data indicate the utilization of different measles vaccines over the years, including the MoPaRu (MMR by Institute of Immunology of Zagreb) vaccine from 1963 to 2007, as well as the Priorix and GlaxoSmithKline MMR vaccines alongside the Croatian national product. Notably, adverse events linked to the mumps component of the MoPaRu (Institute of Immunology of Zagreb) prompted the replacement of the vaccine used for the first dose in 2009, followed by a replacement for the second dose in 2011 due to the discontinuation of production.

In the cohort aged ~30–40 years in Croatia, born between 1993 and 1983 (±1 year), the challenge of addressing the deficiency in humoral antibody levels is pronounced, resonating with experiences observed in Hungary. Previous analyses of Hungarian serum samples have revealed a parallel trajectory [35,66,67,71]. Our prior research endeavors have sought to delineate the potential determinants of vaccine inadequacy. Drawing from national data, we have pursued two main avenues to discern the underlying factors contributing to suboptimal humoral antibody levels [35,66,67,95], recognizing that such insufficiency does not necessarily equate to a complete absence of protection, given the involvement of T cell memory, albeit raising pertinent inquiries. The occurrence of measles epidemics in both examined countries lends support to the plausibility of primary vaccine failure (characterized by the inability to seroconvert post-vaccination) and secondary vaccine failures (manifested by waning immunity following seroconversion) [72,73,76,77,78,79,86,87,88].

When examining the cohort of approximately 20–30-year-old individuals in Croatia, born between 2003 and 1993 (±1 year) (Table 4), the observation of humoral antibody levels significantly diverging from expected standards can be particularly noteworthy. While the primary focus of this analysis lies in epidemiological inquiry rather than delving into the broader realms of political, economic, cultural and social contexts, it is imperative to acknowledge significant historical geopolitical events, like the Yugoslav Civil Wars spanning from 1991 to 2001. These conflicts also exerted notable influence within the Baranja Region in Eastern Croatia, which serves as the primary area for our biological sampling. Understanding the impact of these conflicts might provide valuable contextual insight for our epidemiological investigation. The extensive political instabilities experienced during this period may have resulted in disrupted healthcare systems and hindered access to quality healthcare services and effective response measures for the local population. Therefore, encountering suboptimal sero-epidemiological data within this young age group may be less surprising, given the lasting repercussions of these historical challenges. This presumption may also extend to the previously mentioned cohort of individuals aged 30–40 years. Notably, when examining data pertaining to mumps and rubella alongside measles, a consistent trend is observed across all three measured parameters within these sample cohorts.

Focusing on the Hungarian findings, our current comprehensive analysis aligns with the trends observed in our previous publications [35,66,67,71]. Notably, individuals within the relatively young age group of 30–40 years exhibit a predominant association with suboptimal seroprevalence outcomes. Specifically, within this age cohort, susceptibility to measles and mumps is notably pronounced, with similarly elevated vulnerability observed in relation to rubella. The potential underlying factors have been previously elucidated in our works and are briefly summarized here.

In Hungary, the epidemiological dynamics are presumed to be as follows (Table 3): Prior to the introduction of measles immunization in 1969, measles was endemic, resulting in widespread childhood infection and subsequent long-lasting immunity for most individuals. However, with the advent of immunization and subsequent adjustments to national vaccination policies and schedules, the disease epidemiology underwent significant transformations. Despite these efforts, large-scale measles epidemics persisted, indicating potential deficiencies in the immunization protocols [72,74,75,96]. The epidemic patterns in Hungary reveal distinct outbreaks occurring at different intervals among various age groups. Notably, the significant epidemics in 1973–74 and 1980–81 predominantly affected unvaccinated children aged 6–9 years and 7–10 years, respectively. Subsequently, the 1988–89 epidemic primarily impacted individuals aged 17–21 years, a group targeted for vaccination during early mass campaigns [74]. Furthermore, a minor epidemic in 2017 underscored persistent challenges, attributed to latent susceptibility within the domestic population and proximity to measles-endemic regions [71,74]. This episode underscores the ongoing significance of addressing these issues and the importance of heightened awareness and prioritized interventions [35,71].

In our current comparative analysis, we reaffirm the previously articulated similarities in age-related epidemiological dynamics between Croatia and Hungary, as described in our prior publication [35]. Despite variations in vaccination protocols, immunization doses, geopolitical histories and healthcare system statuses, we observe consistent trends in affected and susceptible age clusters. This suggests that the challenge of eradicating vaccine-preventable diseases, notably measles, is multifaceted, influenced by both vaccination routines and the dynamics of antibody evolution and natural decline. We hypothesize that declines in seropositivity ratios may not solely result from primary or secondary vaccine failure but also from the biological dynamics of vaccination and the waning of long-term immunogenicity [97,98,99,100]. As population immunity increases through vaccination and natural boosting exposures become less frequent, the risk of outbreaks may rise [32,34,35,81,97,100,101].

In summary, we wish to reiterate the paramount importance of raising awareness. Based on the data presented herein and in alignment with ECDC and WHO directives [37,40,53], we suggest that it is crucial to analyze the factors contributing to insufficient vaccine efficacy to tailor interventions effectively. Comprehensive strategies might also include risk communication, initiatives to enhance awareness and training programs for healthcare providers to enable informed vaccination dialogues [37,53]. Given the challenges and burdens associated with responding to epidemiological crises caused by compliance issues, we advocate for early prevention through screening and proactive awareness-raising as preferable to implementing emergency measures in the midst of a healthcare disaster. Furthermore, identifying potential risk groups is essential to enable targeted community-based interventions in the event of virus spread, thereby bolstering vaccination efforts [37,53].

## 5. Conclusions

Measles epidemics have persisted for centuries, but the current surge differs from historical outbreaks due to high vaccination rates. Despite this, outbreaks still occur, posing challenges for healthcare professionals and epidemiologists [3,8,21,22,23,24,25,26,27,28,29,30,31,32,33,34,35,36]. Our age-stratified study, leveraging cross-border sero-epidemiological assessments [35,66,67,71,76], highlights concerns about humoral antibody protection. Suboptimal seropositivity ratios reveal age-specific risk profiles in the cross-border regions of both Croatia and Hungary, potentially identifying high-risk cohorts. Relying solely on favorable vaccination rates and presumed herd immunity can be misleading and undermine vaccination efforts, despite these diseases nearing eradication. Sero-epidemiological screening and monitoring remain crucial, given alarming age-related seronegativity data, especially for measles [37,56,61,62,63,64,65]. Our analysis emphasizes considering factors beyond primary and secondary vaccine failure, noting similarities in trends and seronegativity overlap between countries with varied immunization backgrounds. The dynamics of vaccination, including waning long-term immunity, warrant attention [32,34,35,81,97,98,99,100,101]. Prolonged, structured immunization suppresses epidemic fluctuations and eliminates seasonal virus circulation’s booster effect on immunity, likely causing a decline in vaccine-induced antibody titers [12,37,38,55,62,71,76,81,82,100,101]. Aligned with WHO reports and ECDC directives, preventive measures should address and close immunity gaps, enhance surveillance quality, elaborate outbreak control scenarios and improve compliance in vulnerable settings to bolster vaccination efforts [37,53]. Our analysis aims to underscore the sensitivity of this issue and contribute to assessing targeted interventions and enhancing societal awareness.

## 6. Implications of the Study

In evaluating anti-MMR humoral antibody titers (IgG), distinctions between solely vaccine-induced, solely infection-induced (measles, mumps, or rubella wild-type viruses) and vaccine plus infection-induced “hybrid or combined immunity” cases were avoided due to limitations in specimen nature (clinical residual anonymous sera). The vaccine type and regimen were deduced from birth dates.

This analysis primarily focuses on circulating humoral antibody (IgG) measurement; however, it is noted that cellular immunity assessment was not included, potentially rendering the picture incomplete. Nevertheless, following Plotkin’s definitions, humoral antibody-associated seropositivity ratios can be considered valid “correlates of protection”. Plotkin posits that vaccines primarily function through antibodies that block infection, providing a “correlate of protection”. The functional characteristics and quantity of antibodies are also crucial, as they may correlate highly with protection or synergize with other functions. Immune memory is a critical correlate as well. Some vaccines lack true correlates, relying solely on useful surrogates for an unknown protective response [25,61,81,102,103,104].

Furthermore, there might be a bias in the selection of participants. As mentioned in the materials and methods section, anonymous clinical residual samples were used from the Hungarian partner, specifically from the Clinical Center of the University of Pécs, and from the Croatian partner, specifically from the Scientific Centre for Excellence for Personalized Health Care. The known data regarding the samples included gender and age, without personal names or patient-specific identification numbers. Therefore, in the present study, immunization time points and vaccine regimens can only be deduced based on dates of birth (age cluster-based analysis). Additionally, it is important to note that although multicenter studies have advantages over single-center studies by including a larger number of participants and allowing a more comprehensive population-level examination, they may introduce complexities in data analysis—especially in the case of neighboring countries [105]. The potential drawbacks of sample heterogeneity might also lead to increased variability in the data, complicating the interpretation of the results. Furthermore, this heterogeneity can contribute to confounding bias by encompassing unmeasured or uncontrolled factors [106,107]. Moreover, the use of clinical residual samples [108,109,110] entails the inherent limitation of “loss-of-follow-up”, which, in our case, is the major reason for the infeasibility of cellular investigations targeting T cell memory responses. This limitation results in incomplete data and potential bias. Nevertheless, we believe that our robust sample numbers are sufficient to balance such deviations, and the overlaps found with former reports and similar studies support the reliability of the dataset. Additionally, a highly diversified sample multitude best represents the general immunity status and makeup of a population. Therefore, our samples may truly mimic real-life conditions, rendering our serum bank suitable for representative epidemiological studies.

## Figures and Tables

**Figure 1 vaccines-12-00486-f001:**
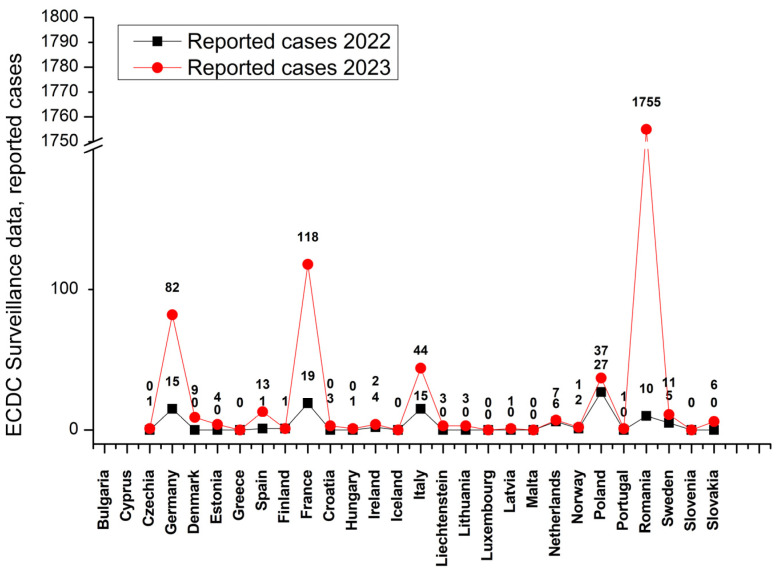
The data depicted in the graphs are sourced from the atlas.ecdc.europa.eu website. The depicted information does not constitute part of the intellectual property of our research group and is solely intended to provide an informative illustration of the serious nature of the problem and to raise awareness. Germany, France, and Romania exhibit the most prominent peaks of increased sample numbers from the year 2022 to 2023. Data source: https://atlas.ecdc.europa.eu/public/index.aspx (accessed on 20 April 2024).

**Figure 2 vaccines-12-00486-f002:**
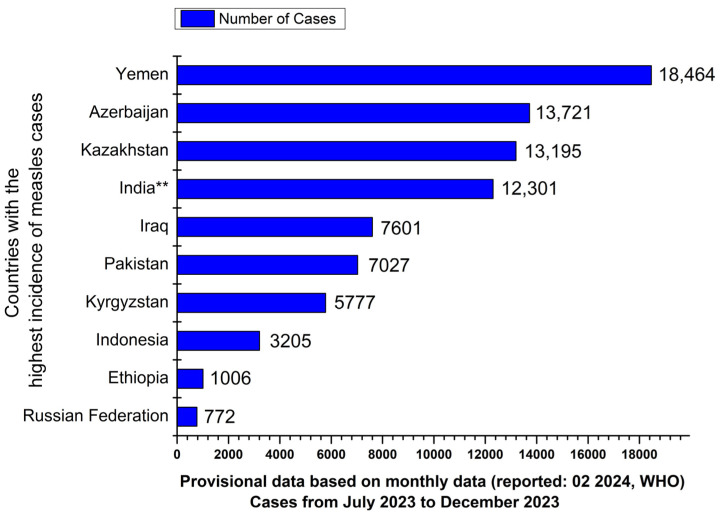
The data depicted in the graphs are sourced from the www.cdc.gov website. The depicted information does not constitute part of the intellectual property of our research group and is solely intended to provide an informative illustration of the serious nature of the problem and to raise awareness. Countries with the highest number of measles cases during the specified period are based on provisional data derived from monthly reports submitted to the World Health Organization (WHO) headquarters in Geneva as of early February 2024. The data cover a time period ranging from July 2023 to December 2023. ** The World Health Organization (WHO) classifies all suspected measles cases reported from India as clinically compatible if a specimen was not collected according to the algorithm for the classification of suspected measles in the WHO Vaccine-Preventable Diseases (VPD) Surveillance Standards. Consequently, there might be disparities between the numbers reported by the WHO and those reported by India. Data source: https://www.cdc.gov/globalhealth/measles/data/global-measles-outbreaks.html#print (accessed on 20 April 2024).

**Figure 3 vaccines-12-00486-f003:**
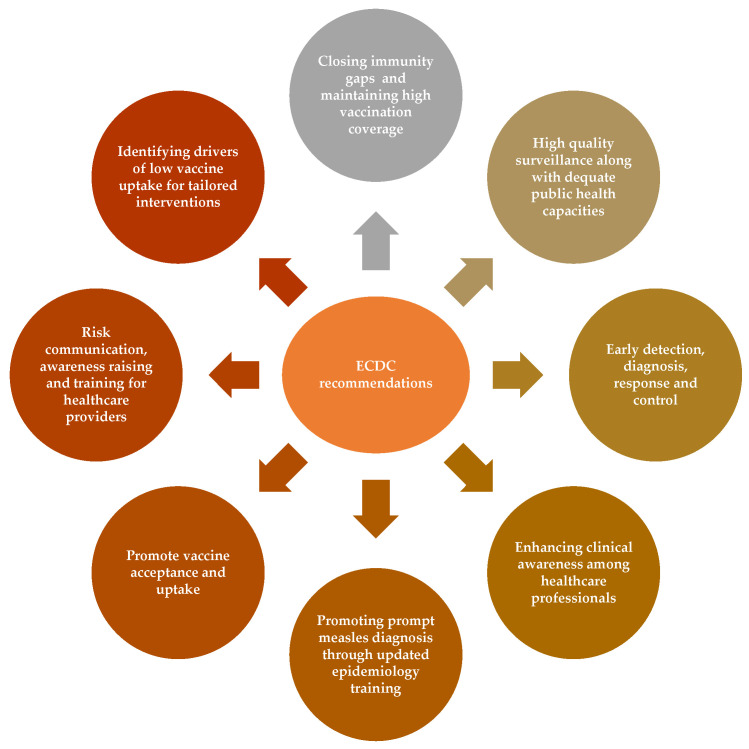
ECDC recommendations for EU/EEA public health authorities regarding an anticipated increase in measles cases. The data shown in the figure are adopted from the ecdc.europa.eu website (THREAT ASSESSMENT BRIEF, Measles on the rise in the EU/EEA: considerations for public health response, 16 February 2024 [37]). The depicted information does not constitute part of the intellectual property of our research group and is solely intended to provide an informative illustration of the serious nature of the problem and to raise awareness. Data source: https://www.ecdc.europa.eu/sites/default/files/documents/measles-eu-threat-assessment-brief-february-2024.pdf (accessed on 20 April 2024).

**Figure 4 vaccines-12-00486-f004:**
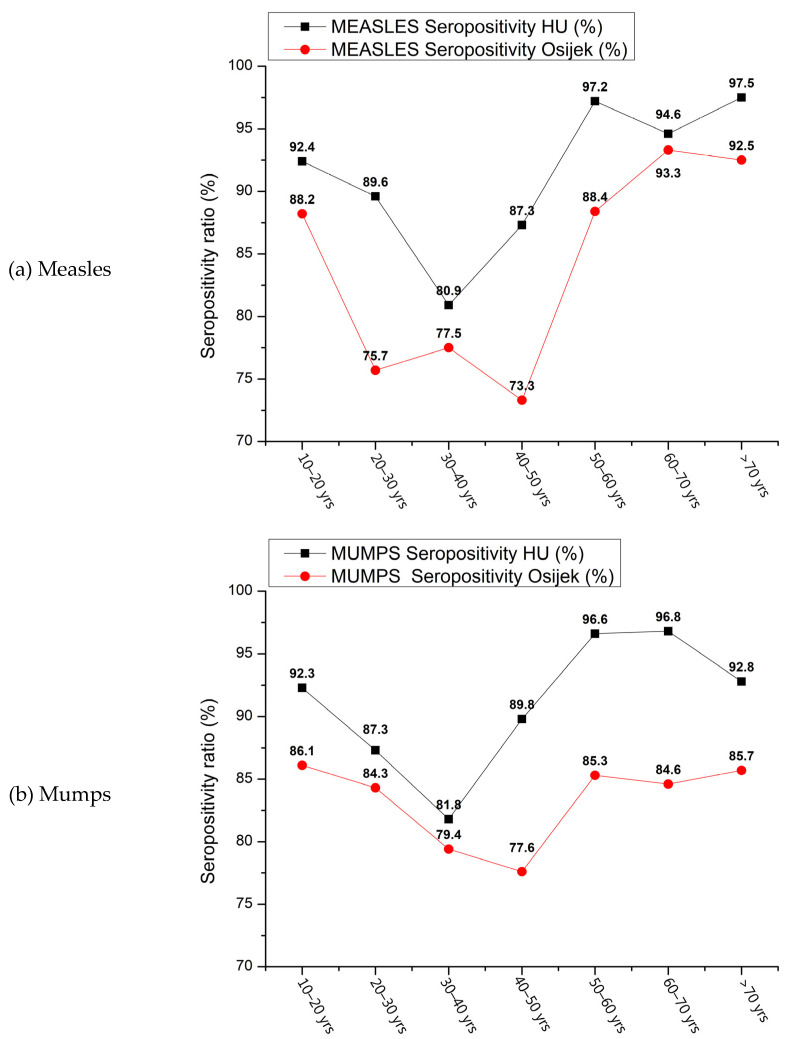
(**a**–**c**). Simple dot plot representation of anti-measles IgG seropositivity ratios. During result computation, no discrimination was made between immune responses elicited by vaccination and those triggered by natural infection, including subsequent seroconversion. The analysis relied on quantitative data obtained through immunoassays (ELISAs), which were then translated into qualitative outcomes (positive or negative). Seropositivity = number of positive samples per age group/total number of samples per age group × 100. Dots are connected solely to facilitate visual tracking of consistent trends between countries, serving the purpose of facilitating easy interpretation based on direct observation. (**a**) Anti-measles IgG seropositivity ratios; (**b**) anti-mumps IgG seropositivity ratios; (**c**) anti-rubella IgG seropositivity ratios.

**Figure 5 vaccines-12-00486-f005:**
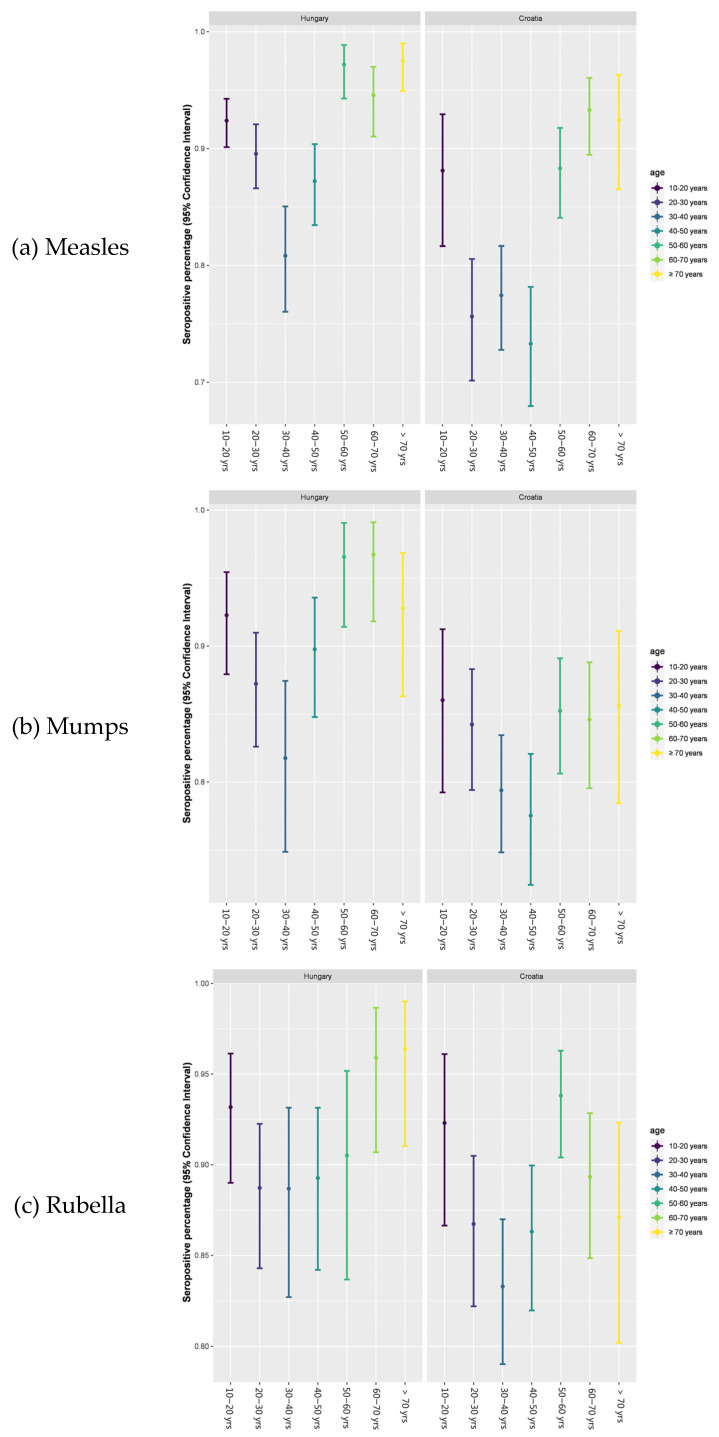
(**a**–**c**). Assessing nonoverlapping confidence intervals for age group comparison. The Clopper–Pearson exact binomial confidence interval was used as a statistical method in order to calculate the confidence interval for a proportion regarding our binomial data (positive or negative seropositivity outcomes). The absence of overlap in confidence intervals (CI 95%) was interpreted as indicative of a statistically significant difference between the respective age groups. For detailed analysis, structured in a clear, table-based design, highlighting only the nonoverlapping confidence intervals between age groups, please see Appendix A; for all computed Clopper–Pearson exact binomial confidence intervals, see Appendix A.

**Table 1 vaccines-12-00486-t001:** Measles, mumps, and rubella sample numbers per age group of the Hungarian and Croatian samples.

		10–20 yrs	20–30 yrs	30–40 yrs	40–50 yrs	50–60 yrs	60–70 yrs	≥70 yrs	Total
Measles	Hungary (Pécs)	682	517	313	383	248	257	280	2680
	Croatia (Osijek)	143	279	359	307	291	253	132	1764
Mumps	Hungary (Pécs)	220	266	159	205	116	122	111	1199
	Croatia (Osijek)	143	279	359	307	291	253	132	1764
Rubella	Hungary (Pécs)	220	266	159	205	116	122	111	1199
	Croatia (Osijek)	143	279	359	307	291	253	132	1764

**Table 2 vaccines-12-00486-t002:** Measles, mumps and rubella. Number of seronegative samples versus total sample number per age group.

		AGE (Years)	10–20	20–30	30–40	40–50	50–60	60–70	≥70	Total
Measles	Hungary (Pécs)	Sample numbers per age group	682	517	313	383	248	257	280	2680
		Number of seronegative samples	52	54	60	49	7	14	7	284.39
	Croatia (Osijek)	Sample numbers per age group	143	279	359	307	291	253	132	1764
		Number of seronegative samples	17	68	81	82	34	17	10	309
Mumps	Hungary (Pécs)	Sample numbers per age group	220	266	159	205	116	122	111	1199
		Number of seronegative samples	17	34	29	21	4	4	8	117
	Croatia (Osijek)	Sample numbers per age group	143	279	359	307	291	253	132	1764
		Number of seronegative samples	20	44	74	69	43	39	19	308
Rubella	Hungary (Pécs)	Sample numbers per age group	220	266	159	205	116	122	111	1199
		Number of seronegative samples	15	30	18	22	11	5	4	105
	Croatia (Osijek)	Sample numbers per age group	143	279	359	307	291	253	132	1764
		Number of seronegative samples	11	37	60	42	18	27	17	212

**Table 3 vaccines-12-00486-t003:** Measles/MMR vaccination schedules in Hungary.

Year/Period of Vaccination	Who Received Vaccinations This Year, and What Were the Underlying Rationales for Their Administration?
Prior to 1969	Patients who have not received vaccinations are susceptible to wild-type infections or have been through a wild-type virus infection. In 1969, the measles vaccine was introduced in Hungary, utilizing the live, attenuated Leningrad-16 strain manufactured in the Soviet Union.
1969–1977	Between 1969 and 1974, a single dose of the measles vaccine was administered during widespread campaigns to individuals aged 9–27 months. Initially, the recommended age for vaccination was 10 months, until it was adjusted to 14 months in 1978. After an initial decline in the incidence rate, notable epidemics emerged, predominantly among unvaccinated children aged 6 to 9 years, during the period spanning 1973–1974. Following the epidemic of 1980–81, individuals born from 1973 to 1977, who would have been vaccinated at 10 months, were given a revaccination. The 1988–89 epidemic predominantly affected individuals aged 17–21 years, who were prioritized for vaccination during the early phases of the vaccination program in Hungary. Subsequently, starting in 1989, children were routinely revaccinated at the age of 11 with the monovalent measles vaccine according to a structured schedule. As a result, the earliest recipients of this 11-year reminder vaccination were born in 1978. Consequently, the cohort born between 1969 and 1977 represents the final group not included in the official vaccination schedule to receive a reminder vaccine at age 11.
1978–1987	These are the first individuals who benefited from the reminder monovalent measles vaccine at the age of 11. In 1999, the administration of the trivalent vaccine was started in Hungary; consequently, those who received the first trivalent vaccine in 1999 were born in 1988.
1988–1990	In 1989, the rubella vaccine was introduced, coinciding with the initiation of the monovalent measles reminder vaccination at the age of 11. The following year, in 1990, the measles–rubella bivalent vaccines were introduced.
1991–1995	The initiation of the initial vaccine administration at 14 months of age persisted from 1978 until 1991. In 1991, the measles–mumps–rubella (MMR) trivalent vaccine was introduced. Subsequently, in 1992, the MMR vaccine was administered at 15 months of age. The MERCK MMR II, featuring the Enders’ Edmonston strain (live, attenuated), was introduced in 1996.
1996–1998	In 1996, the MERCK MMR II, incorporating the Enders’ Edmonston strain (live, attenuated), was introduced. In 1999, a shift occurred from the monovalent measles vaccine to the measles–mumps–rubella (MMR) revaccination. This transition coincided with the introduction of GSK PLUSERIX, featuring the Measles Schwarz Strain.
1999–2002	In 1999, the GSK PLUSERIX vaccine, containing the Measles Schwarz Strain, was introduced. Subsequently, in 2003, the GSK PRIORIX vaccine was introduced.
2003	In 2003, the GSK PRIORIX vaccine, containing attenuated Schwarz Measles, was introduced.
2004–2005	During the years 2004 to 2005, the MERCK MMR II vaccine was administered.
2006–2010	From 2006 to 2010, during a five-year tender period, the GSK PRIORIX vaccine containing attenuated Schwarz Measles was utilized.
After 2011	Starting in 2011, a Sanofi-MSD product, MMRvaxPro, containing the live attenuated Measles virus Enders’ Edmonston strain, has been employed for both the initial vaccination and revaccination of children. Meanwhile, GSK PRIORIX remains available on the market and is predominantly utilized for vaccination in adulthood.

Sources of information include International Notes Measles—Hungary, MMWR Weekly, October 06, 1989/38(39); 665–668 [73], relevant national and international public sites on vaccination calendars, information, and safety [90,91], as well as verbal information obtained from partners and colleagues at the Hungarian National Institute of Epidemiology (with special regards to Dr. Zita Rigó and Dr. Zsuzsanna Molnar).

**Table 4 vaccines-12-00486-t004:** Measles/MMR vaccination schedules in Croatia.

Vaccination Period (Years)	Who Received Vaccinations This Year, and What Were the Underlying Rationales for Their Administration?
…–1968/69	1968: The measles vaccine was incorporated into the national childhood vaccination schedule [90]. During the initial phases of vaccine implementation, individuals with a history of prior measles infection were generally not targeted for vaccination. Diagnosis primarily depended on medical history and clinical presentation, leading to the possibility that some children were not immunized due to underrecognition of past infections.
1968–1969	The live measles vaccine was cultivated in human diploid cells (WI-38) at the Institute of Immunology of Zagreb from a further-attenuated Edmonston–Zagreb strain originally propagated in tissue culture in chick embryos. The Edmonston–Zagreb strain of measles virus is a further-attenuated Edmonston–Enders strain that has undergone 19 passages in human diploid cells (WI-38), including three plaquings [92]. Based on contemporary data, post-immunization reactions induced by the Edmonston–Zagreb vaccine were categorized as mild. The incidence of individuals experiencing fever exceeding 38 °C was less than 2%. Additionally, a fourfold rise in antibody titers among the seronegative cohort exceeded 90% [92].
1969	The implementation of large-scale measles vaccination initiatives began in the former Yugoslavia in 1969, utilizing a monovalent measles vaccine for both primary and booster doses. Children born between 1965 and 1967 who had not contracted the measles virus (MeV) were targeted for vaccination. Additionally, children attending first grade during the 1968/69 school year (typically aged 6 or 7, born in 1962 or 1963) and who remained free from measles infection were included in the vaccination campaign. Immunization efforts extended to infants in their eleventh month of life. Furthermore, children scheduled for vaccination in 1968 (those born in 1966), as well as subsequent cohorts, including second-grade students (aged 7 or 8, born in 1961–1962), and those in childcare facilities who missed vaccination opportunities due to various reasons, were also prioritized for immunization.
1970	Children born between 1963 and 1968 who had not been previously exposed to measles and had not undergone any vaccination were administered immunization, except for those designated to receive the third dose of the DTaP (Diphtheria, Tetanus, Pertussis) vaccine. Additionally, vaccination was provided to children in the fourth grade of elementary school during the 1969/70 academic year (aged 9 or 10; born in 1959 or 1960), who had not encountered the measles virus and had not yet received vaccination. Furthermore, infants in their eleventh month of life were administered vaccination following the continuous protocol.
1973	Primary vaccination was administered to children at one year of age, with the additional inclusion of the rubella component.
1974	The mumps component of the vaccine was added
1975	1975: The rubella vaccine introduced in the national childhood vaccination schedule [90]. In 1975, children older than one year who followed a consistent vaccination schedule were set to receive their initial vaccination. Moreover, children born in 1973 eligible for targeted vaccination campaigns, excluding those awaiting their third DTaP dose, were designated to receive their first vaccination. Additionally, children over one year of age enrolled in preschool facilities who had not yet been vaccinated were also scheduled for their initial vaccination. Furthermore, children born in 1971 and those entering first grade in the 1974/75 academic year were also slated to receive their initial vaccination.
	In 1976, the MMR trivalent vaccine was officially integrated into the national childhood vaccination schedule, replacing single-antigen vaccines for the first dose and introducing a mumps vaccination program. Additionally, a rubella catch-up vaccination program for 14-year-old girls was initiated in the same year [90]. In 1976, the Institute of Immunology in Zagreb introduced a trivalent measles–mumps–rubella vaccine, replacing the monovalent vaccine used for the initial dose. As a result, children received their first trivalent vaccinations against measles, mumps, and rubella (MMR) through ongoing vaccination protocols beginning after their first year of life since that time. Under the campaign vaccination approach, all children born in 1974, except those set to receive the third dose of DTaP (Diphtheria, Tetanus, Pertussis) during that timeframe, received their first vaccinations against measles, mumps, and rubella (MMR). Additionally, girls in the eighth grade of elementary school (born in 1963 or 1962) received their initial rubella vaccination. Furthermore, children entering first grade during the 1975/76 school year (aged 6 or 7, born in 1970 or 1969) received the measles vaccination.
1994	In 1994, a second dose of MMR (MMR2) was introduced at 7 years of age, replacing the single-antigen vaccines for the second dose [90]. Since 1994, the trivalent vaccine of the Institute of Immunology in Zagreb has been routinely utilized for the administration of the second dose as well.
1996	Children who, for any reason, did not receive their initial MMR vaccination remained eligible for vaccination up to the age of 14. Additionally, all girls attending eighth grade during the 1996/97 academic year (aged 13 or 14, born in 1983 or 1982) received the rubella vaccination. The present regulations prohibit exemptions from vaccination for individuals who have previously experienced measles, mumps, or rubella infections.
1997	Since 1997, it had been recommended to administer MMR2 at 12 years of age [90]. The timing for revaccination, initially slated for administration during the first grade of elementary school, had been adjusted to take place in the sixth grade.
1999	In 1999, the recommendation for MMR2 was reverted back to 7 years of age [90].
2008–2009	PRIORIX (GlaxoSmithKline), a live attenuated combined vaccine against measles, mumps, and rubella, is recommended for active immunization against these infections. PRIORIX is a lyophilized mixed preparation of the attenuated Schwarz measles, RIT4385 mumps (derived from the Jeryl Lynn strain) and Wistar RA 27/3 rubella strains of viruses, separately obtained by propagation either in chick embryo tissue cultures (mumps and measles) or MRC5 human diploid cells (rubella). In pediatric settings, a single dose is typically advised for children, either on or shortly after their first birthday. Older children lacking documented evidence of prior vaccination should also receive the vaccine [93].
2009	Due to adverse events caused by the mumps component of the national ‘MoPaRU’ (MMR) vaccine (produced by the Institute of Immunology in Zagreb), which occurred after the first dose of the vaccine, this vaccine was replaced for the first dose by another producer in 2009. (Due to the discontinuation of its production in 2011, this vaccine was replaced by another, also for the second dose.)
2010	The aforementioned PRIORIX (GlaxoSmithKline) and M-M-RVaxPro (Merck Sharp & Dohme) are two commercially available vaccines used to confer protection against measles, mumps and rubella in individuals aged 12 months or older. M-M-RVaxPro may be administered to infants between 9 and 12 months of age under specific circumstances [94]. This vaccine contains live attenuated strains of measles virus (Enders’ Edmonston strain), mumps virus (Jeryl Lynn [Level B] strain) and rubella virus (Wistar RA 27/3 strain) [94]. In addition to these commercial products, the national vaccine “MoPaRU” (MMR), produced by the Institute of Immunology in Zagreb, remained in use until 2011.
2011–2014	PRIORIX (GlaxoSmithKline) and M-M-RVaxPro (Merck Sharp & Dohme)
2015–…	PRIORIX (GlaxoSmithKline)

Information relies on co-author accounts, literature overview [76,77,79,90,92] and commercial vaccine product inserts [93,94].

## Data Availability

Research data and investigation results are available upon request.

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
