# Peer review of "Raising Epidemiological Awareness: Assessment of Measles/MMR Susceptibility in Highly Vaccinated Clusters within the Hungarian and Croatian Population—A Sero-Surveillance Analysis"

_vaccines, 2024, doi:10.3390/vaccines12050486_

Round 1
Reviewer 1 Report
Comments and Suggestions for Authors
In this paper, the author addresses the issue of decreased measles vaccination rates during the COVID-19 pandemic, which heightened the risk of outbreaks even in adequately vaccinated populations by age stratification method.
This article is timely and as someone currently working on measles prevalence in the USA with my colleagues, I find the results and conclusion of this study very interesting.
Major comments
The introduction should capture the recent spike in measles cases in the USA, which is also due to low vaccination rates.
I think this study should capture the number of doses taken by the population considered. The authors should study the number of doses.
This study is largely descriptive in methodology and I strongly think that there are other statistical methods the authors can use or develop that will go in line with the scope of their study.
Minor comments
I don't understand 1983 to 1973 in line 289, perhaps it is a mistake.
Tables 2 and 3 should move to the appendix. They are too long.
I observed some spelling errors. An example can be found in lines 164 and others.
I can't find Table X that the authors were referring to in line 293.
Comments on the Quality of English LanguageMinor editing of English language required
Author Response
Dear Reviewer,
We are grateful for the opportunity to submit a revised version of the manuscript titled "Suboptimal Measles/MMR Seropositivity Rates in Highly Vaccinated Populations - Elevating Epidemiological Awareness." We sincerely appreciate the time and thoroughness with which you reviewed our manuscript. Your insightful comments have been invaluable in enhancing the quality and rigor of our work.
We have conscientiously attended to the suggestions and concerns articulated by the reviewers, implementing appropriate revisions to the manuscript accordingly. Throughout this process, we have tracked and documented all changes made, ensuring transparency and accountability in our revisions.
Once again, we extend our gratitude for your constructive feedback, which has undoubtedly strengthened the manuscript. We are confident that the revised version now better aligns with the standards of scientific rigor and contributes meaningfully to the field of epidemiology.
Sincerely,
Authors
Comments and answers - Reviewer 1
In this paper, the author addresses the issue of decreased measles vaccination rates during the COVID-19 pandemic, which heightened the risk of outbreaks even in adequately vaccinated populations by age stratification method.
This article is timely and as someone currently working on measles prevalence in the USA with my colleagues, I find the results and conclusion of this study very interesting.
Major comments
The introduction should capture the recent spike in measles cases in the USA, which is also due to low vaccination rates. – Thank you very much for your insightful comment. A relevant paragraph has been included in the introduction section, starting with “Moreover, in January 2024, the WHO Region of the Americas…” (line 151- 161). Furthermore, the pertinent communication from the Pan American Health Organization (PAHO) and World Health Organization (WHO) is cited multiple times within the manuscript.
I think this study should capture the number of doses taken by the population considered. The authors should study the number of doses. - Thank you for your valuable remark. Due to the anonymized nature of the samples, detailed patient immunization records supported by clinical histories were not available. The manuscript acknowledges the implications of utilizing anonymous residual clinical samples on multiple occasions. Nevertheless, efforts have been made to provide a thorough contextualization of vaccination schedules relevant to the examined countries, particularly highlighting age-related complexities. Table 3 contains data on the number of vaccine doses administered to the population in Hungary, while Table 4 presents the corresponding information for Croatia. These tables offer a comprehensive overview of the modifications made to vaccination schedules, including the timing of primary and booster immunizations, targeted demographics, administration techniques, and the types and formulations of the administered vaccines. By delineating age group boundaries within the aforementioned tables, connections between vaccination and epidemiological trends become traceable and interpretable. Upon your comment, we added an explanatory paragraph to the Discussion session between lines 379-387.
This study is largely descriptive in methodology and I strongly think that there are other statistical methods the authors can use or develop that will go in line with the scope of their study.
Thank you for your insightful suggestion. Following your recommendation, we opted to employ non-overlapping confidence intervals as a method to depict differences between groups. Clopper-Pearson exact binomial confidence interval was used as a statistical method in order to calculate the confidence interval for a proportion regarding our binomial data (positive or negative seropositivity outcomes). Absence of overlap in confidence intervals (CI 95%) was interpreted as indicative of a statistically significant difference between the respective age groups. Yet, we refrained from providing p-values as a measure of statistical significance (where p < 0.05 indicates rejection of the null hypothesis) due to variances stemming from differences in case numbers. In the context of analyzing numerous variables, reliance solely on the p-value may not represent the most robust statistical approach. We would like to highlight the importance of considering False Discovery Rate (FDR) Correction in such scenarios. FDR Correction is essential when conducting multiple statistical tests, such as comparisons across various groups or variables, as it addresses the challenge of increased likelihood of observing a 'significant' result by random chance alone, known as the multiple comparisons problem.
Following your recommendation, Figure 5 has been incorporated into the Results section, accompanied by relevant explanations delineating its significance within the context of the study findings.
Minor comments
I don't understand 1983 to 1973 in line 289, perhaps it is a mistake. - Thank you very much for your important comment. We have indeed found that this information was redundant here; therefore, it has been removed.
Tables 2 and 3 should move to the appendix. They are too long. - Thank you for this valuable insight. We have indeed been deliberating on the placement of these extensive tables from the outset. Your comment confirms that they are too voluminous for the Introduction section, as they also contain descriptive historical data explaining the rationale behind specific targeted vaccination efforts. Following your suggestion, we have now relocated these tables to the Discussion section. However, we find it necessary to keep them accessible within the main text, as certain parts of the Discussion refer to their content. We believe this arrangement enhances reader convenience.
I observed some spelling errors. An example can be found in lines 164 and others. -Thank you for the alert. Spelling errors have been carefully revised throughout the entire text.
I can't find Table X that the authors were referring to in line 293.- Thank you. The error has been rectified.

Reviewer 2 Report
Comments and Suggestions for Authors
The study investigates suboptimal seropositivity rates for MMR in populations from Hungary and Croatia despite high vaccination coverage. The paper explores the discrepancies in immunity levels, particularly in adult cohorts, and assesses the risk factors and implications of such immunity gaps in supposedly well-immunized communities.
Strengths:
1. The study is based on a robust dataset of 2,680 Hungarian and 1,764 Croatian serum samples, providing a substantial basis for the conclusions drawn.
2. By comparing two distinct populations, the study provides valuable insights into the regional variations in MMR immunity, enriching the understanding of seroepidemiological dynamics.
3. Analyzing age-stratified immunity profiles helps identify specific age groups at higher risk, which is crucial for targeted public health interventions.
Drawbacks:
1. The paper does not adequately address the potential biases or limitations inherent in the study design, such as selection bias or the representativeness of the sample populations.
2. While the paper mentions the influence of socioeconomic and geopolitical factors on vaccination rates and immunity, it does not provide a detailed analysis of these aspects or their direct correlation with the observed seropositivity rates.
3. The focus is predominantly on IgG levels to define seropositivity. The paper could benefit from a more nuanced discussion on the role of cellular immunity and other aspects of immunological memory, which are also critical for long-term protection against infections.
Recommendations:
1. Incorporating a more comprehensive analysis of socioeconomic and cultural factors could provide deeper insights into the barriers to achieving optimal vaccination coverage and immunity.
2. Adding assessments of cellular immune responses could provide a more complete picture of the population’s immune status against MMR, particularly in the context of understanding long-term immunity and vaccine effectiveness.
3. Given the findings, the paper could extend its implications to specific public health policy recommendations, particularly in terms of revising vaccination strategies and public health messaging to address the identified immunity gaps.
Author Response
Dear Reviewer,
We are grateful for the opportunity to submit a revised version of the manuscript titled "Suboptimal Measles/MMR Seropositivity Rates in Highly Vaccinated Populations - Elevating Epidemiological Awareness." We sincerely appreciate the time and thoroughness with which you reviewed our manuscript. Your insightful comments have been invaluable in enhancing the quality and rigor of our work.
We have conscientiously attended to the suggestions and concerns articulated by the reviewers, implementing appropriate revisions to the manuscript accordingly. Throughout this process, we have tracked and documented all changes made, ensuring transparency and accountability in our revisions.
Once again, we extend our gratitude for your constructive feedback, which has undoubtedly strengthened the manuscript. We are confident that the revised version now better aligns with the standards of scientific rigor and contributes meaningfully to the field of epidemiology.
Sincerely,
Authors
Comments and answers - Reviewer 2
The study investigates suboptimal seropositivity rates for MMR in populations from Hungary and Croatia despite high vaccination coverage. The paper explores the discrepancies in immunity levels, particularly in adult cohorts, and assesses the risk factors and implications of such immunity gaps in supposedly well-immunized communities.
Strengths:
- The study is based on a robust dataset of 2,680 Hungarian and 1,764 Croatian serum samples, providing a substantial basis for the conclusions drawn.
- By comparing two distinct populations, the study provides valuable insights into the regional variations in MMR immunity, enriching the understanding of seroepidemiological dynamics.
- Analyzing age-stratified immunity profiles helps identify specific age groups at higher risk, which is crucial for targeted public health interventions.
Drawbacks:
- The paper does not adequately address the potential biases or limitations inherent in the study design, such as selection bias or the representativeness of the sample populations - Thank you very much for the valuable suggestion. Following your comment, we have expanded the 'Implications of the Study' section with a more detailed discussion on selection bias, the inherent limitations associated with using clinical residual samples, the advantages and disadvantages of multicenter sample banking, and the resulting potential bias in result interpretation. We acknowledge that these aspects were previously omitted and are crucial additions for a more thorough understanding by the readers of the manuscript.
- While the paper mentions the influence of socioeconomic and geopolitical factors on vaccination rates and immunity, it does not provide a detailed analysis of these aspects or their direct correlation with the observed seropositivity rates.- Thank you for your valuable input. We appreciate your attention to detail. Please refer to our response provided for Recommendation number 1. Additionally, we wish to underscore the significant implications associated with utilizing anonymous residual clinical samples, a limitation that has been acknowledged multiple times throughout the manuscript. The anonymity of the samples (Implications of the study) precluded the determination of key socioeconomic determinants, such as income, occupation, education, and place of residence, and the potential resulting socioeconomic inequalities (SES). While our Discussion section briefly touches upon the impact of the 'Yugoslav Civil Wars' on vaccine uptake, we have refrained from further exploration of this sensitive topic. This decision is primarily rooted in two considerations: Firstly, the scope of our manuscript primarily centers on epidemiological inquiry rather than delving into broader political, economic, cultural, and social contexts. Secondly, we recognize and respect the potential personal sensitivities of our Croatian partners regarding this matter. Upon your suggestion, we have added 2 paragraphs to the Introduction section, detailing Socioeconomic inequalities’ (SES) constitute pivotal determinants influencing vaccine uptake, and implications for national economies and healthcare caused by the currently ongoing mass migration phenomenon (lines 53-66).
- The focus is predominantly on IgG levels to define seropositivity. The paper could benefit from a more nuanced discussion on the role of cellular immunity and other aspects of immunological memory, which are also critical for long-term protection against infections. populations - Thank you for the important remark. An important limitation of the manuscript is our focus solely on humoral antibody titers, as already discussed in the 'Implications of the Study' section. However, according to Plotkin's definitions, humoral antibody-associated seropositivity ratios can be considered valid "correlates of protection." On your suggestion, we have highlighted this limitation once more (lines 564-567). This limitation is also tied to the inherent implication of working with a serum bank of clinical residual samples, which includes the limitation of "loss-of-follow-up." In our case, this limitation prevented cellular investigations targeting T cell memory responses due to the inability to recall patients for a second blood draw, as we did not have their names on record. To address this in future studies, we plan to obtain extended ethical permissions. These permissions would allow us to gather information about both humoral and cellular immune responses from the same individuals. This approach will certainly offer a more comprehensive understanding of the post-vaccination (or post-infection) immune responses.
Recommendations:
- Incorporating a more comprehensive analysis of socioeconomic and cultural factors could provide deeper insights into the barriers to achieving optimal vaccination coverage and immunity. – We appreciate your valuable insight. Following your suggestion, we have expanded the Introduction section by incorporating two additional paragraphs (lines 53-66) that delve into the significant role of socioeconomic inequalities (SES) as key determinants impacting vaccine uptake. Furthermore, we have mentioned the implications for national economies and healthcare systems resulting from the ongoing mass migration phenomenon.
- Adding assessments of cellular immune responses could provide a more complete picture of the population’s immune status against MMR, particularly in the context of understanding long-term immunity and vaccine effectiveness. – Thank you for this valuable insight. Please see the answers given to ‘Drawbacks’ nr 1 and nr 3.
- Given the findings, the paper could extend its implications to specific public health policy recommendations, particularly in terms of revising vaccination strategies and public health messaging to address the identified immunity gaps. – Thank you for your valuable comment. In response to your suggestion, we have included 2 additional paragraphs (lines 151-161) that delve into the epidemiological concerns arising from the current situation in the Pan-American region. We have also incorporated PAHO/WHO recommendations alongside pertinent suggestions and guidance. Regarding preventive intervention measures in Europe, directives are included between lines 162-169.
Moreover, at your suggestion, a schematic figure illustrating preventive intervention scenarios, based on ECDC's 'Threat Assessment Brief' of February 2024 has been created (Figure 3). This figure has been added to the introduction section, with the aim of providing a clearer understanding to potential readers of our alignment with ECDC's recommendations and strategic factors.
Additionally, in the Discussion section, we reference multiple times the February 2024 'Threat Assessment Brief' (ECDC), underscoring that our article was crafted in alignment with the respective directives (Aligned with the directives set forth by the ECDC, our current article highlights key points emphasized in the February 2024 'Threat Assessment Brief.' - starting in line 360).

Round 2
Reviewer 1 Report
Comments and Suggestions for Authors
I am satisfied with the authors response.